# Digital Health Support: Current Status and Future Development for Enhancing Dialysis Patient Care and Empowering Patients

**DOI:** 10.3390/toxins16050211

**Published:** 2024-04-30

**Authors:** Bernard Canaud, Andrew Davenport, Hélène Leray-Moragues, Marion Morena-Carrere, Jean Paul Cristol, Jeroen Kooman, Peter Kotanko

**Affiliations:** 1School of Medicine, Montpellier University, 9 Rue des Carmelites, 34090 Montpellier, France; 2Fondation Charles Mion, AIDER-SANTE, 34000 Montpellier, France; h.leray@aidersante.com (H.L.-M.);; 3MTX Consulting International, 34090 Montpellier, France; 4UCL Department of Renal Medicine, University College London, London WC1E 6BT, UK; a.davenport@ucl.ac.uk; 5PhyMedExp, Department of Biochemistry and Hormonology, INSERM, CNRS, University Hospital Center of Montpellier, University of Montpellier, 34000 Montpellier, France; m-morenacarrere@chu-montpellier.fr; 6Department of Internal Medicine, Division of Nephrology, Maastricht University Medical Center, 6202 AZ Maastricht, The Netherlands; 7Renal Research Institute, Icahn University, New York, NY 10065, USA; peter.kotanko@rriny.com

**Keywords:** end-stage kidney disease, kidney replacement therapy, patient outcomes, digital technology, artificial intelligence, uremic toxins, cost-effectiveness

## Abstract

Chronic kidney disease poses a growing global health concern, as an increasing number of patients progress to end-stage kidney disease requiring kidney replacement therapy, presenting various challenges including shortage of care givers and cost-related issues. In this narrative essay, we explore innovative strategies based on in-depth literature analysis that may help healthcare systems face these challenges, with a focus on digital health technologies (DHTs), to enhance removal and ensure better control of broader spectrum of uremic toxins, to optimize resources, improve care and outcomes, and empower patients. Therefore, alternative strategies, such as self-care dialysis, home-based dialysis with the support of teledialysis, need to be developed. Managing ESKD requires an improvement in patient management, emphasizing patient education, caregiver knowledge, and robust digital support systems. The solution involves leveraging DHTs to automate HD, implement automated algorithm-driven controlled HD, remotely monitor patients, provide health education, and enable caregivers with data-driven decision-making. These technologies, including artificial intelligence, aim to enhance care quality, reduce practice variations, and improve treatment outcomes whilst supporting personalized kidney replacement therapy. This narrative essay offers an update on currently available digital health technologies used in the management of HD patients and envisions future technologies that, through digital solutions, potentially empower patients and will more effectively support their HD treatments.

## 1. Introduction: Challenges in the Management of End-Stage Kidney Disease Patients and Opportunities for Digital Health Support

Chronic kidney disease (CKD) has emerged as a significant contributor to non-communicable diseases and mortality in the 21st century [1,2,3,4]. This is partly due to increased risk factors including aging, diabetes mellitus, hypertension, and vascular disease. This trend is expected to result in a higher number of CKD patients, leading to an increased incidence of patients progressing to end-stage kidney disease (ESKD) and requiring kidney replacement therapy [2,3,5].

This increasing number of ESKD patients poses a series of challenges, not only in terms of availability of dialysis resources and trained staff but also along with escalating health care costs [6]. An additional challenge arises from the changing patient profile, as healthcare professionals encounter more complex problems with increasing combinations of risk factors: aging, comorbidities, and varying degrees of disabilities. Current clinical guidelines are based on historic patient demographics and clinical practices, but a dynamic adaptation of kidney replacement therapy is also required to meet changing patient needs and risks, so personalizing treatment to provide both an effective healthcare system and improve patient coping with kidney replacement therapy (KRT) burden [6,7]. Despite recent improvements in the treatment of ESKD patients, global studies have emphasized that CKD patients and dialysis treatment are leading causes of cardiovascular mortality and overall poor patient health-related quality of life worldwide [8,9,10,11].

This escalating burden on healthcare systems, driven by the increasing number of CKD patients progressing to ESKD, poses substantial challenges [5]. Containment strategies involve innovations in healthcare delivery, resource optimization, and proactive management approaches [7]. Rapid advancements in dialysis, especially in understanding uremic toxins and adopting new therapeutic approaches, necessitate healthcare professionals staying informed to provide state-of-the-art care to CKD patients. Addressing the shortage of medical professionals and caregivers is critical for optimal CKD patient care. Attracting and retaining healthcare professionals, streamlining workflows, and embracing telemedicine and digital health are key to tackling healthcare challenges [12,13].

Managing ESKD thus requires a paradigm shift in treatment approaches, including anticipating and addressing complications, adjusting treatment plans, and optimizing patient care for better long-term outcomes. Overcoming this challenge involves implementing strategies to enhance the overall treatment efficiency of kidney replacement therapy, ensuring better control of uremic toxins levels, improving patient education and perception, updating caregivers’ knowledge, and establishing robust digital support systems to positively influence overall results [2,7,14].

In this developing landscape, digital health support emerges as a potential pivotal solution, leveraging technology to support caregivers with innovative solutions, including automation and self-adaptation of dialysis machines equipped with algorithms responding to user-specific prescriptions, continuously monitoring quality and controlling care delivery, with teledialysis or teleconsultation options, or remote patient monitoring with pervasive tools, health education platforms, and data-driven decision-making [15,16]. This evolving digital health environment with artificial intelligence support will enable healthcare providers to improve patient care, reduce practice variations, and effectively manage the complexities associated with ESKD, and so potentially improve treatment outcomes.

Digital health technologies (DHTs), particularly to support the management of patients with chronic diseases, are not a new concept [17,18]. Digital solutions have long been considered potential and cost-effective ways of delivering aspects of healthcare, reducing practice variations in care delivery, enhancing global treatment efficiency and uremic control levels, and ultimately improving patient outcomes while reducing overall treatment costs. In this narrative, we will focus on the use of digital solutions and tools to support the care of ESKD patients on maintenance dialysis.

## 2. Potential of Digital Health Technologies in Dialysis Patient Care

DHTs currently encompass a broad spectrum of tools for dialysis patient care. These tools fall into several different categories. Firstly, there are technical functionalities integrated into hemodialysis (HD) machines designed to enhance, secure, and improve daily patient care. Secondly, data is automatically collected and analyzed from various sources, including the dialysis machine, weighing scale, vital parameters (including blood pressure, heart rate, temperature), clinician electronic medical records, laboratory results, imaging reports, and medications. Thirdly, remote tools utilize wearable sensor devices such as wristwatches, electronic scales, oximeters, sleep disorder monitors, and physical activity trackers. Figure 1 illustrates a schematic representation that integrates digital health technology with the HD patient.

In essence, these tools offer potentially valuable information that can be utilized at the point of care to facilitate patient management and identify corrective actions. When patients are at home, they can use this information to provide insights or trigger specific actions. Additionally, these tools can be integrated into feedback-controlled algorithms that react in real time to data, implementing preset corrective actions. This represents the initial and fundamental level of support that can be provided by digital health technology. Table 1 provides a concise summary of the main tools intended to optimize care delivery, empower patients, and ultimately improve patient outcomes and reduce treatment burden.

### 2.1. Tools Enabling Care Delivery and Improving Outcomes

Furthermore, these devices can be connected to the internet and web systems, thus allowing transmission of data to a cloud-based system capable of storing, analyzing, processing, and providing more extensive support to both patients and clinicians. Such an approach necessitates the support of advanced analytics and artificial intelligence, as information collected from a large dataset of patients can be employed as a quality control to check that selected indicators are on target, to develop predictive medicine, and/or assist decision-making at an individual level.

HD machines currently incorporate basic monitoring features that ensure safety of treatment and provide valuable information for point-of-care use in patient management and follow-up [19]. While these tools can help identify corrective actions, they necessitate caregiver intervention and should be regarded as fundamental components of HD machines relying on activation of safety alarms [20].

Advancements in DHT are leading to more sophisticated tools being integrated into advanced HD machines, which will be discussed in the sections below. These newer tools are equipped with feedback-controlled algorithms capable of reacting to changes in physiologic parameters (i.e., blood volume or temperature change) and implementing preset corrective actions almost immediately [21,22]. This initial level of automation and self-adaptation represents the foundational support provided by digital health technology, which is already implemented as an optional function in the current generation of advanced HD machines [23,24].

A subsequent level of advancement, currently in development, involves highly sophisticated tools that interface the dialysis machine, point-of-care information, and patient electronic medical records with a cloud-based advanced analytics system using web-based technology [25,26,27]. This second level aims to provide decision-making support to physicians and caregivers, refining prescriptions or enhancing the delivery of dialysis [28,29].

To categorize the initial level of automation and self-adaptation, it is convenient to group them into three main categories based on their intended therapeutic actions. Firstly, tools that are designed to enhance hemodynamic tolerance, primarily aiming to prevent intradialytic hypotension caused by volume imbalances. Secondly, there are tools dedicated to improving treatment efficiency and facilitating the delivery of optimal care. Lastly, tools aimed at preventing resource wastage, specifically focusing on reducing water consumption.

#### 2.1.1. Feedback-Controlled Tools Designed to Improve Hemodynamic Tolerance

A feedback-controlled system refers to an algorithm that senses physiological changes in a patient’s biomarker during HD (HD) and provides an adapted response capable of correcting this change and mitigating the side effects associated with it. Hemodynamic instability is the most common adverse event in HD, mainly related to the hypovolemia induced by ultrafiltration rate, resulting in intradialytic hypotension, ischemic events, and intradialytic morbidity. Therefore, improving hemodynamic stability has been the focus of investigation for several years. In addition to increasing treatment time or frequency to reduce the ultrafiltration rate, there are three main approaches to improving hemodynamic stability in intermittent short HD patients employing feedback-controlled algorithms: the first approach relies on volume-controlled conditions by acting either on ultrafiltration or dialysate sodium or a combination of both; the second approach focuses on primarily increasing vascular resistance and venous capacitance by controlling thermal balance; the third approach consists of sodium management by automatically aligning dialysate sodium to plasma sodium. It is worth noting that these approaches may be used independently or in combination. These three approaches are further discussed below.

A.Ultrafiltration Controlled System with Profiles

To maintain volemia within a user-defined corridor and prevent critical hypovolemia based on the patient profile and hemodynamic response, a fuzzy control system is employed. This system utilizes analog input values that range continuously between 0 and 1 to modulate the ultrafiltration and vascular refilling rates, thereby ensuring that volemia remains precisely within a safe range [23,25,26,30,31]. Essentially, a sensor on the arterial blood line continuously measures hematocrit (using ultrasound density or refractometry). Changes in hematocrit concentration, attributed to net ultrafiltration (resulting in hemoconcentration), provide data to a central processing unit. This unit translates information into the relative blood volume change (expressed as a percentage) or vascular refilling rate and feeds this into an algorithm, which adds a correction for the Fahraeus effect, due to differences in hematocrit in different vascular beds. This algorithm, in turn, then controls the ultrafiltration rate to maintain volemic changes within the safety corridor set by the user [32,33].

Moreover, various algorithms, referred to as profiles, may be applied to facilitate tissue fluid recruitment and vascular refilling, preventing the occurrence of critical hypovolemia and occurrence of intradialytic hypotension or ischemic event [34]. Typically, these profiles combine ultrafiltration rate and dialysate sodium concentration, starting with a higher ultrafiltration rate and hypertonic dialysate that gradually decreases throughout the session, concluding with a lower ultrafiltration rate and isotonic dialysate. The challenge lies in ensuring adequate removal of both sodium and water mass to prevent long-term sodium and fluid overload.

Several studies have indicated that such ultrafiltration-controlled systems can significantly reduce intradialytic hypotension, improve hemodynamic and overall tolerance during dialysis sessions, and mitigate the burden on patients and caregivers associated with coping better with KRT [35,36]. For example, in a recent review, fatigue associated with dialysis was proportionally related to interdialytic weight gain and inversely to the ultrafiltration rate, confirming that acting on this parameter is crucial [37]. However, as of now, no study has demonstrated that this approach can improve long term cardiovascular outcomes.

B.Thermal Balance Controlled System

HD sessions conducted with a fixed dialysate temperature of 37.5 °C are known to result in positive thermal balance, indicating that patients gain significant thermal energy during the session [38]. This hyperthermic dialysis is attributed to most patients starting with a core temperature close to 36.5 °C. The combination of this warming process and higher ultrafiltration rate can adversely affect the hemodynamic response to hypovolemia, leading to vasodilation and tachycardia, aggravating hemodynamic instability in fragile patients [39,40].

Numerous studies, including meta-analyses, have confirmed that implementing isothermic or discretely hypothermic dialysis sessions significantly improves hemodynamic tolerance. This improvement is demonstrated by a reduced incidence of intradialytic hypotension and an increase in mean arterial pressure during dialysis sessions [39,41,42]. Additionally, clinical studies have demonstrated that hypothermic dialysis has a cardiac and vascular protective effect, particularly on the brain, as demonstrated in an interventional study using sophisticated MRI imaging of the heart and the brain [43,44,45,46,47]. In essence, isothermic or hypothermic dialysis sessions are highly preferable for enhancing hemodynamic tolerance and preventing organ damage caused by dialysis-induced systemic stress [48,49].

However, the recent MyTemp study conducted in Canada did not corroborate previously reported findings [50,51]. This study, however, had notable flaws in the assessment of hemodynamic stability Additionally, it used a much lower dialysate temperature than previous studies, which was very close to the patient’s core temperature, providing already an isothermic dialysis condition. In essence, this study could not provide a definitive answer to the question of the role of cooling dialysate [51,52].

In this context, manual or, preferably, automated thermal balance adjustment based on the individual patient’s core temperature, using specific blood temperature sensors and adapted algorithm, is preferred to alleviate shivering effects resulting from arbitrary reductions in dialysate temperature.

C.Automated Sodium Controlled Systems

Various tools utilizing conductivity sensors as surrogates for sodium concentration, along with specific proprietary algorithms, have now been developed to optimize sodium and water imbalance management for HD patients [53,54,55]. A recent review by Petitclerc et al. offers technical insights and clinical applications of these devices. In essence, there are two main categories of sodium management tools with proprietary algorithms [56,57]. The first relies on estimating plasma sodium, either using ultrafiltrate (Mozarc Medical, Minneapolis, MN, USA) or an equilibrated dialysate to plasma concentrations achieved by recirculation (Baxter, Deerfield, MA, USA) at the beginning of the dialysis session. The second relies on measuring dialysate sodium concentrations (inlet and outlet) based on conductivity measurement corrected for potassium changes to ensure a perfect sodium mass balance and to estimate the initial plasma sodium concentration using ionic clearance (Fresenius Medical Care, Bad Homburg, Germany). An important distinction lies in the intended use of these tools. In the first group, the user sets a final plasma sodium target, and the algorithm adjusts dialysate sodium while calculating sodium mass removal. In the second group, the user sets a dialysis-plasma sodium gradient, initially set by default isotonic to plasma and termed “zero diffusive”. Based on patient conditions, this gradient can be set differently, either negatively (e.g., −3 mmol/L) or positively (e.g., +3 mmol/L), to alter sodium mass removal and/or plasma tonicity changes. In both cases, sodium mass balance is estimated across the dialysis session, but in the second option, diffusive and convective sodium mass removal are differentiated.

#### 2.1.2. Tools Designed to Enhance Treatment Efficiency and to Optimize Care Delivery

A.Online Clearance Measurement of Dialysis Efficiency

Online clearance measurement of dialysis efficiency is essential for ensuring optimal treatment outcomes for HD patients [58]. Continuous monitoring of solute clearances, particularly urea clearance or its surrogate, ionic dialysance, is facilitated by online tools integrated into dialysis machines. These tools provide the user a means of correcting deviations from the targeted prescription in real time by adjusting the dialysis prescription (i.e., blood flow, treatment time). This cost-effective method provides a reliable means to assess treatment efficiency across all dialysis sessions and allows for prescription adjustments in response to variations during treatment delivery (e.g., changes in blood flow, recirculation, thrombosis, or treatment time) [59,60].

These measurements rely on internal algorithms and proprietary technology within the dialysis machine, utilizing various sensors. One type employs conductivity probes (inlet and outlet) with timely pulses in dialysate conductivity [58,59,60,61], while another uses a UV adsorption sensing chamber on the outlet dialysate, measuring changes in absorptiometry at specific wavelengths to determine dialysate concentrations of the solute of interest [62]. Notably, the UV absorptiometry approach potentially allows for dosing various solutes based on their specific wavelength characteristics, including urea, creatinine, phosphate, or beta-2 microglobulin [62,63,64,65,66,67,68].

These online clearance measurement tools are integrated into various dialysis machines, providing invaluable immediate point-of-care information to caregivers about the efficiency of individual dialysis treatments [62,69]. The DOPPS study has demonstrated that regular delivery of effective dialysis treatments is a key factor in improving outcomes for dialysis patients [70]. Online tools face practical concerns in daily practice: firstly, they are not regularly used by caregivers in their workflow processes to check and improve practices, indicating a need for implementation of training; secondly, while they have been proven to be accurate, reliable, and valid in clinical research settings, their widespread adoption in daily routines might be hindered by economic restrictions. In essence, while technology has the potential to offer invaluable support in quantifying and personalizing therapy, the critical challenge lies in determining who will bear the cost of implementing these advanced technologies [59].

B.Sodium and Water Management

Optimal management of sodium and water balance, known as sodium homeostasis, is recognized as essential for improving outcomes in dialysis patients, particularly regarding cardiac health [71]. The traditional clinical approach consists of the dry weight probing approach [72]. The safety of this isolated clinical approach has been questioned, while benefits of instrumental guidance (i.e., bioimpedance, lung ultrasound) or cardiac biomarkers have been regularly highlighted [73,74,75]. In this context, the availability of automated sodium-controlled tools has renewed clinical interest in improving outcomes for dialysis patients [71,76]. Several studies have confirmed the validity and reliability of these sophisticated tools for managing sodium and water imbalances during HD sessions, including high-volume hemodiafiltration (HDF) patients [77,78]. Preliminary short-term studies have also confirmed the reliability of their use. For example, in active sodium management mode, aiming isonatremic (isotonic) dialysis condition, changes in plasma sodium concentration during HD sessions were less than 1.0 mmol/L and closely resembled those observed during isolated ultrafiltration [54]. These tools provide regular estimates of sodium mass balance (total and differentiated convective and diffusive) and identification of hyponatremic patients, and have shown benefits in addressing intradialytic morbidities including headaches, paradoxical hypertension, and fatigue in relatively recent studies [76,77,78]. Moreover, they tend to reduce interdialytic weight gain by alleviating thirst induced by osmotic changes [77,78]. However, long-term studies are still required to confirm longer term cardiovascular health benefits.

C.Automated Substitution and Ultrafiltration Control in Hemodiafiltration

Automated substitution control in HDF is essential to ensure matching of the targeted substitution and convective volumes while preventing the transmembrane pressure alarms due to membrane fouling. This technology is incorporated into the HDF monitor, sensing viscosity changes within the hemofilter during the HDF session and adjusting substitution and ultrafiltration flows to maintain the transmembrane pressure within an ideal range. Depending on the type of HDF monitor (i.e., Baxter^®^, Mozarc Medical^®^, Fresenius Medical Care^®^), the management of transmembrane pressure relies on different algorithms [24,79,80]. Notably, the most sophisticated, namely AutoSub^+^ from Fresenius Medical Care, allows a gradual increase in transmembrane pressure during the initial phase of HDF [81,82,83]. This ensures slow fouling of the membrane, beneficial for preventing albumin loss, followed by stabilizing the transmembrane pressure within an optimal range between 150 and 300 mmHg [84,85,86].

In all cases, it has been demonstrated that such automated ultrafiltration control increases the total substitution and ultrafiltration volume delivered over a 4 h HDF by 4 to 5 L, while preventing activation of disruptive transmembrane pressure alarms [81,82,83,87].

D.More Sophisticated Biosensors Are Currently in Development or Being Tested.

Access to biological fluids, such as plasma or dialysate, offers an easy opportunity for the non-invasive measurement of blood constituents. This can be employed for real-time monitoring of the kinetic clearance of key solutes, evaluating the efficiency or adequacy of dialysis s (e.g., uremic compounds, urea, creatinine, B2M, indoxyl sulfate) [67,68,88] or facilitating feedback control of electrolytes (sodium, potassium, bicarbonate, calcium) using appropriate microchip biosensors [89].

#### 2.1.3. Tools Designed to Promote More Sustainable Dialysis

Both HD and on-line HDF use large volumes of water, and water consumption has environmental consequences. Dialysate and water consumption in both HD and HDF depend on the user’s prescription settings and the operational capabilities of the HD/HDF machines. As such, manufacturers have developed different strategies to reduce water consumption. Firstly, with HDF, the substitution flow is drawn from the total dialysis fluid flow without compensation, thereby reducing the dialysis fluid passing through the dialyzer by the same amount. Secondly, the dialysis fluid and substitution flows are independently set, ensuring that the dialysis fluid passing through the dialyzer remains at a constant flow unaffected by the substitution flow. An alternative approach is that the HDF machine can manually or automatically align the dialysis fluid flow to the blood flow (Qd/Qb) to optimize solute saturation and dialysate consumption. The dialysate to blood flow ratio may be set between 1.5 in HD and 1.2 in online HDF. When addressing concerns about dialysis fluid and water consumption in HD and HDF, it is crucial to precisely match these settings with the specific type of HD or HDF machine [90]. The clinical benefits of high-dose HDF has been demonstrated in the CONVINCE study [91], and the higher sustainability of HDF compared to HD in this context has been shown in a recent study, which we encourage readers to delve into for further details [92].

#### 2.1.4. Tools for Interfacing Dialysis Machines and Monitoring Devices in a HD Unit or Network System

An integrated medical informatics system in a dialysis unit plays a pivotal role in facilitating and securing the workflow of a patient, from their arrival at the HD center to their departure [93]. This relies on a meticulously designed electronic network system that encompasses servers and caregiver tools (laptops, tablets). The system connects and interfaces dialysis machines, monitoring devices (such as weight scales and automated blood pressure monitors), patient electronic medical records, and various internal services via an intranet [94,95]. These internal services include pharmacy, integrated laboratory and imaging results, administration with secretarial, and billing services. Additionally, it interfaces with external services through the internet, covering healthcare services, insurance, transportation, general practitioners, and specialists. These solutions exemplify the convergence of healthcare and information technology (IT) systems, representing the culmination of a long-term evolving field known as medical informatics [96,97].

However, development of a single medical informatics system is frequently impeded by the multitude of medical devices within a dialysis unit and their lack of interoperability, making it challenging to share data with IT systems without manual intervention. The pressing need to improve medical care, optimize workflow, eliminate paper forms, reduce costs, and embrace electronic medical records (EMRs) compels clinics, hospitals, and care providers to adopt IT solutions [98]. These IT solutions are designed to streamline various procedures, encompassing electronic medical records processing, quality control, laboratory testing, medical imaging, and administrative tasks including correspondence, transportation, and billing [96]. Despite the complexity and potential cost of implementing such global medical informatics, the adoption of such common and communicating IT systems will reduce and reward healthcare costs in the long term.

To overcome communication barriers between medical devices and IT networks, several digital operators’ groups have developed and proposed integrated medical informatics systems including conversion units to deliver such services. Such solutions share the common objective of converting medical device data into electronic medical records (EMRs) and transmitting them to private cloud services. This enables the performance of data analytics to enhance the assessment of a patient’s condition, with the capability to report dashboard results [95].

### 2.2. Empowering Patients through Digital Health Support

Empowering dialysis patients has been demonstrated to significantly enhance their perception of treatment, instill confidence and self-worth, improve mood, and alleviate anxiety or depression, ultimately leading to improved overall outcomes while enhancing and reducing peri-dialytic fatigue and other symptoms [99,100]. Empowering dialysis patients through digital health tool support can manifest in various forms [101,102], briefly described here.

#### 2.2.1. Facilitating Self-Care Empowerment and Home Treatment Options

In-center HD remains the most frequently used option for several reasons, despite the potential for more flexible, efficient, comfortable, and cost-effective therapeutic options with home HD [99,100]. However, a barrier to home HD is the patient’s perception of isolation without direct medical assistance. Digital tools and IT systems, including tablets or digital tools integrated into or alongside dialysis machines, can significantly assist patients in staying connected with their reference center [102]. These tools facilitate communication and enable patients to benefit from direct and continuous assistance, breaking the isolation associated with home treatment and restoring self-confidence and trust in dialysis through teleassistance [100,101,102].

#### 2.2.2. Educating and Coaching Patients

Digital connected personal tools (such as tablets and cellular phones) have the potential to provide valuable information to dialysis patients, not only regarding their conditions and educational tool but also by sharing annotated results of their dialysis treatment, including clinical, biological, or paraclinical testing. Pervasive remote tools (such as watches, trackers, and vital sensors) as part of enhancing well-being may provide details on physical activity, estimated caloric expenditure, metabolic data, sleep quality, and vital parameters. Furthermore, these digital tools may offer guidance on the timing of medication administration and reactivate functionalities to assist patients in more effectively managing their health.

#### 2.2.3. Monitoring Patient Remotely and Detecting Potential Life-Threatening Complications

Recognizing that mortality among HD patients often occurs during the interdialytic period, especially after longer intervals (e.g., 3 days), and is frequently associated with cardiovascular causes, addressing the heightened incidence of arrhythmias becomes crucial [103,104,105]. This increased risk has been unequivocally identified through implantable monitoring devices, revealing a prevalence much higher than previously reported. Implementing vigilant monitoring for ambulatory dialysis patients is essential to identify those at risk at an earlier stage.

Utilizing currently available pervasive remote tools, such as connected watches (e.g., iWatch, ScanWatch), is hypothesized to aid in the early identification of arrhythmic disorders or critical sleep apnea, facilitating timely corrective actions [101]. The optimal utilization of remote monitoring or tracking tools, including weight scales, blood pressure monitors, heart rate monitors, sleep monitors, oxygen saturation monitors, and temperature sensors connected to a cloud, would be immensely beneficial in monitoring and managing high-risk patients, as discussed in a recent comprehensive review [15,101].

#### 2.2.4. Assessing Patient Reported Outcomes in Real-Time

Patient-self-reported outcome measures (PROMs) are increasingly vital tools for evaluating health-related quality of life (HRQoL) and the burden associated with KRT. Current practices in HD involve the use of questionnaires such as the SF36, KDQoL, and EQ5 to assess and monitor health-related quality of life and to gauge the treatment adequacy of HD patients [106]. While these tools demonstrate significant predictive value for patient outcomes, they are somewhat cumbersome to administer and lack high specificity or sensitivity in detecting subtle changes in dialysis patients [107,108,109].

Recent advancements have introduced digitally supported tools, such as PROMIS^®^, designed to assess patient-reported outcome measures more efficiently [110]. As previously mentioned, the CONVINCE study has developed and tested a web-based, kidney-specific questionnaire linked to a dedicated library with computer-adaptive questions [91]. These questions delve more specifically into various domains of dialysis patients, utilizing a tablet interface. This innovative approach is poised to usher in a new era in the assessment of PROMs in dialysis patients.

## 3. Future Developments and Trends

Additional tools, currently available or in development stage, leverage digital health support to facilitate and enhance care delivery or transition towards personalized dialysis treatment aligned with precision medicine. They are briefly outlined below and schematically presented in Figure 2.

### 3.1. Tools Designed to Facilitate Self-Care, Limited Care, and Remote Care Support for HD Patients, Including Teledialysis or Telemedicine

Among the challenges in providing effective kidney replacement therapy are the shortage of caregivers and the increasing cost of treatment. To partially address these challenges, it has been demonstrated that implementing self-care, limited care, and remote care supported by teledialysis is an important option. This approach ensures the maintenance of quality care while containing costs. Several experiments, particularly in countries with large remote areas (e.g., Canada, Australia, France), have been conducted worldwide with conclusive positive results [111].

In this context, digital health support technologies can play a crucial role in ensuring the safety and success of dialysis therapy. Without delving into details, we refer interested readers to guidance and best practices that have been developed to support the implementation of such approaches [112,113,114,115,116].

### 3.2. Tools of Continuous Quality Improvement in Patient Management

The DOPPS study clearly demonstrated that implementation of best practice patterns in HD, including achieving treatment time and dialysis dose, as well as vascular access policy and education and training of caregivers, significantly influence patient outcomes, often more so than the technical or dialysis therapy options [117,118]. Policy makers and caregivers should be aware of this finding, emphasizing that ensuring the correct and optimal delivery of care as prescribed is the most effective way to reduce mortality for HD patients, as well as the correct of caregiver-to-patient ratio. This discovery underscores the importance of continuous quality improvement in dialysis care and adequate staffing and competence as cornerstones of success.

Given the complexity of managing dialysis patients and the multitude of parameters or data, covering various domains, to consider (such as clinical parameters, vital signs, subjective symptoms, dialysis machine parameters, laboratory values, and imaging data), it is evident that only an electronic medical record capturing, analyzing, and presenting simplified dashboards with visual graphic data can provide adequate and timely support to physicians and caregivers [95,96].

The use of digital tools, along with advanced analytics and imaging tools presented via a visual and user-friendly dashboard, has shown tremendous value in a large chain of dialysis care providers, where optimal efficiency is a top priority. This technology improves patient outcomes by identifying individuals or groups of patients deviating from targeted key parameters, enabling swift corrective action.

As evidenced in a recent prospective study within a large international network of dialysis care providers, the combination of monthly quality control assessments clustered in various categories of interest (i.e., dialysis dose, fluid and hemodynamic control, vascular access, electrolytes, bone mineral disorders, nutritional, inflammatory, and anemia control) associated with peer review significantly reduced hospitalization and mortality rates [119]. This approach streamlines the implementation of corrective actions and, consequently, minimizes variations in care delivery practices among different caregivers and units.

Several examples illustrate this purpose, such as increasing dialysis dose delivery by adjusting blood flow and effective treatment time, promoting the increased use of native arteriovenous fistula, reducing erythropoietin use to correct anemia, and improving fluid volume management when combined with bioimpedance devices.

### 3.3. Tools Designed to Support Decision-Making and Advance towards Personalized Dialysis

Predictive medicine is emerging as an innovative and valuable tool to support physicians in their daily practice. Drawing on extensive population data and large datasets, the development of sophisticated predictive models, whether guided or not, is primarily driven by artificial intelligence. These tools offer new opportunities to predict events, outline trajectories of biomarkers, and assess responses to specific therapies or treatment modalities with a high level of reliability.

To illustrate this point, we have identified four categories of tools which are voluntary exhaustive:

#### 3.3.1. The First Category Is Dedicated to Establishing the Risk Stratification Profile of Incident Dialysis Patients and Predicting Their Outcomes within the Next One or Two Years

Various tools within this category have been developed, requiring basic information including patient characteristics, underlying kidney disease, comorbidities, clinical and laboratory data, and the choice of treatment modality [120,121]. By leveraging these elements, predictive tools can forecast patient outcomes over the next one or two years [121]. Simulating the effects of a specific treatment modality on mortality can assist physicians and patients in determining the optimal therapeutic option in these scenarios. As example, such tools may prevent selection biases based on subjective physician assessment and introduce a more objective method with multiparametric items to estimate the chance of success or failure according to dialysis modality choice (i.e., home dialysis versus in-center) before engaging the patient and/or family in a particular treatment modality. Such an approach will reduce patient disease burden and the risk of failure.

#### 3.3.2. The Second Category Aims to Support Physician Decision-Making in Controlling Anemia through Erythropoietin-Stimulating Agents and Iron Supplementation

Recent studies indicate that modeling anemia control using machine learning can correct anemia more efficiently than conventional approaches [122]. This involves maintaining hemoglobin in a narrow range (10 to 12 g/dL), virtually eliminating hemoglobin fluctuations, reducing the consumption of erythropoietin-stimulating agents and iron supplementation, and tending to decrease the occurrence of cardiovascular events.

#### 3.3.3. The Third Category Centers on Predicting the Occurrence of Intradialytic Hypotension within a HD Session [28,29]

Two recent studies have demonstrated that predicting intradialytic hypotension or hemodynamic events was possible within the next 15 to 75 min with an accuracy of 85%, as estimated from the area under the curve (AUC) parameter. This achievement resulted from combining dialysis prescription and patient features, data from previous sessions, and data trends from the current dialysis session. Interestingly, it appears that artificial intelligence utilizing machine learning was more accurate than traditional modeling relying on correlations.

#### 3.3.4. The Fourth Category Focuses on the Allocation of Dialysis Modality to the Most Appropriate Patients to Optimize Their Survival Expectancy [123]

Several tools have already been developed to predict patient outcomes over the next years, with external validation conducted through large databases [123,124,125]. It is important to note that these tools are not designed to replace the clinical judgment of physicians or patient preferences. Instead, they have the capacity to predict outcomes over the next few years with reasonable accuracy and can serve as a valuable support for both physicians and patients in making informed choices.

### 3.4. Tools to Assess Health Related Quality of Life

HD is linked to a substantial symptom burden and diminished health-related quality of life (HRQoL). Patient self-reported outcome measures (PROMs), standardized tools capturing patients’ symptom burden, functional level, and HRQoL, offer a valuable means to monitor overlooked health aspects, guide care planning, and support treatment adherence or choice. The integration of PROMs into clinical practice serves as a strategic empowerment tool for patients, involving them in care decisions and outcomes [107,108,109].

Despite the potential benefits, the incorporation of PROMs faces challenges within busy HD units, characterized by limited human resources and the burden on patients in completing questionnaires. However, the growing demand from health authorities necessitates the implementation of PROMs and their connection to actionable treatment interventions to alleviate symptoms and enhance patient well-being.

Practical considerations include selecting specific questionnaires reflecting CKD patients’ concerns, analyzing reports, and integrating results into electronic medical records for optimal utilization. Here, electronic tools such as tablets, utilizing web-based platforms with adaptive questions based on patient concerns (as exemplified by PROMIS^®^), present a potentially disruptive approach [110].

By leveraging on these self-adaptive and easily handle tools, it is plausible that dialysis patients can regularly provide PROMs, offering caregivers more efficient opportunities for corrective actions or treatments.

## 4. Barriers to Adoption of Digital Health Technologies in Dialysis Patient Care

As with all new and potentially disruptive technologies, there are several barriers to overcome for the implementation and widespread acceptance of digital health technologies [15,124,125]. We will briefly address the main challenges along with potential solutions.

### 4.1. Technical Challenges Are Not the Most Complex to Solve

As indicated, technology progresses rapidly, offering new solutions every day. HD machines and patient environments are already equipped with numerous digital and connected medical devices that allow the capture and transmission of a substantial amount of data to server and cloud systems. The next challenge lies in capturing and analyzing the data, providing actionable information to users to facilitate care and decision support. Subsequently, the challenge will be to securely store big data within an appropriately secured cloud system.

### 4.2. Ethical Challenges Are Certainly More Complex to Address, as They Involve Issues Related to Individual Privacy

Addressing confidentiality and the risk of sharing personal health data is a daunting task, especially considering recent cyberattacks and data breaches. Ensuring the anonymization and secure storage of personal health data requires significant additional effort.

### 4.3. Regulatory Challenges Are Twofold

First, there is the concern related to data protection and individual privacy, falling under the General Data Protection Regulation (GDPR) in the European Union (EU). The GDPR aims to enhance individuals’ control and rights over their personal information. Given the high trade exchange between the EU and the USA, efforts should be made to align EU data protection regulations with international rules. The second regulatory concern involves medical devices, governed by the European Medical Device Regulation. Devices such as HD machines, water treatment systems, and other medical devices, including monitoring and electronic networks, must adhere to specific features outlined in their registration files.

### 4.4. Liability Challenges Represent Another Aspect That Needs Addressing, Particularly as They Involve Various Devices and Manufacturers

While liability is clearly defined for specific medical devices, it becomes more complex when multiple devices interact and interface through an electronic network responding to a cloud-based analytic system.

### 4.5. Personal Challenges May Reflect the Fears and Reluctance of Caregivers, Including Doctors, to Adopt New Medical Devices, Especially Those with New Functionalities Which May Interfere with Their Existing Skills, Competence Domain, or Job Responsibilities [125]

Overcoming these human reflexes requires appropriate learning and training to master the functionalities of the new medical technologies.

## 5. Conclusions

Managing ESKD patients requires a paradigm shift in treatment approaches to address the increasing demand, optimize patient care for improved long-term outcomes, and contain overall costs. In this landscape, digital health technologies emerge as a pivotal solution, leveraging technology to support caregivers with innovative solutions. This digital health support, whether embedded in the dialysis machine or provided through external support via a digital network and artificial intelligence, will potentially enable healthcare providers to enhance patient care, reduce practice variations, improve care quality, effectively manage the complexities associated with ESKD, and optimize treatment outcomes. However, these innovative technologies must overcome barriers related to technical, ethical, regulatory, liability, or personal beliefs for their implementation and widespread acceptance within nephrology communities.

## Figures and Tables

**Figure 1 toxins-16-00211-f001:**
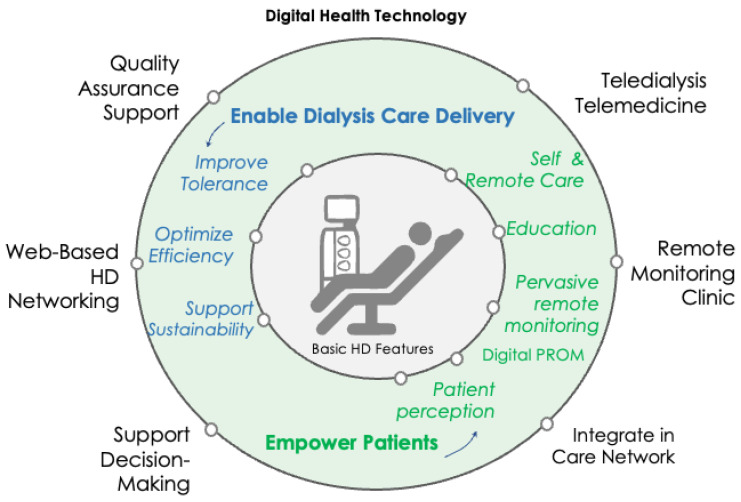
Integrating digital health technology (DHT) in HD: a visual representation. The digital health technology in HD is then conceptually grouped into two main levels: the inner circle indicates patient-level application of DHTs, such as tools that enable dialysis care delivery (blue) and patient self-empowerment (green); the outer circle indicates system-level applications of DHTs aimed to support quality assurance, the implementation of HD networking, and decision-making through artificial intelligence and big data analytics. Abbreviations: HD, hemodialysis; PROM, Patient-Reported Outcome Measures.

**Figure 2 toxins-16-00211-f002:**
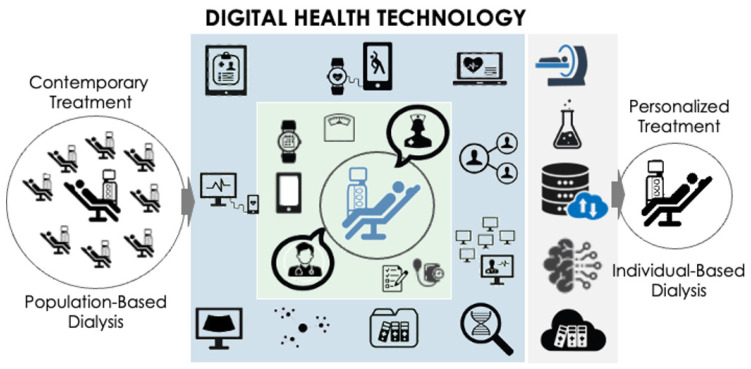
**Digital health technology helps to transition from population-based to individual based dialysis.** Additional tools, currently available or in development, are presented to leverage digital health technology support. These tools aim to facilitate and enhance care delivery, fostering the transition from contemporary population-based dialysis towards personalized and individualized dialysis.

**Table 1 toxins-16-00211-t001:** **Tools supported by digital health technology to support care of chronic kidney disease patients on hemodialysis.** It provides a concise summary of the main tools intended to optimize care delivery, empower patient and ultimately improve patient outcomes and reduce treatment burden.

Category	Tool	Function	Aim
**Tools improving care delivery**	Feedback controlled tools	Ultrafiltration and dialysate sodium Control with Profiles	Enhance hemodynamic management and improve hemodynamic stability and tolerance
		Thermal balance Control	
		Automated Sodium Control	
	Enhance treatment efficiency and optimize care delivery	Online Clearance Measurement of Uremic Compounds (Urea, ß2M)	Ensure continuous quality control and enhance treatment efficiency
		Fluid volume Management Including Sodium and Water	Enhance fluid volume and hemodynamic management
		Automated Substitution and Ultrafiltration Control in online HDF	Enhance dialysis treatment efficiency
		New biosensors on effluent dialysate (IS, PCS, Electrolytes, Na, K, Ca)	Optimize and personalize treatment
	More sustainable dialysis	Automated adjustment of dialysis fluid production and flow to blood flow	Optimize water and electrolyte consumption, protect the planet
	Interfacing HD machines and monitoring devices in a IT network	Integrated medical informatics	Optimize the management of patient care and the resource utilization
**Tools empowering patients**	Self care and home treatment	Adapted and portable HD machines	Improve HR-QOL
	Educating and coaching	Digital connected and remote devices	Personalize and optimize the delivery of treatment
	Patient remote monitoring	Pervasive remote monitoring	Prevent cardiac sudden death
	Patient Reported Outcomes Monitoring	Digital PROMs supported by tablet and AI	At the forefront: patient-centered care

## Data Availability

Data are contained within the article.

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
