# Peer review of "Digital Health Support: Current Status and Future Development for Enhancing Dialysis Patient Care and Empowering Patients"

_toxins, 2024, doi:10.3390/toxins16050211_

Round 1

Reviewer 1 Report

Comments and Suggestions for Authors

This is a comprehensive and rather unique review of digital health technology (DHT) in the field of hemodialysis, covering past accomplishments, ongoing explorations, and future opportunities. This Reviewer has a few comments that may help the authors at the revision stage.

Main comments

In the Abstract and Introduction, outline the method of review, e.g., systematic, scoping; and indicate the field of review, e.g., in-center hemodialysis only, home hemodialysis, hemodialysis and hemodiafiltration, peritoneal dialysis etc.

Regarding sentence in the Abstract and the theme throughout the manuscript: “Managing ESKD requires a shift in treatment paradigms, emphasizing patient education, caregiver knowledge, and robust digital support systems.” Consider rephrasing. This might be viewed as an overstatement. The paradigm of ESKD management should have already included patient and caregiver education: The fact that it has not been done to satisfaction it does not constitute a paradigm shift. However, perhaps DHT will aid in facilitating and enhancing patient and caregiver education.

Introduction, page 1, line 39, consider changing “and satisfy patient perception” to “and improve patient coping with dialysis treatments”. To “satisfy” patient perception appears a few times throughout the manuscript. Consider rewording, e.g., to ‘mitigate patient and caregiver burden in coping with dialysis’ etc

Introduction, page 1, line 42, “and overall poor patient health related quality of life perception worldwide”. Consider removing the word ‘perception’ because poor QoL is a reality rather than just perception.

Figure 1. is missing a title, consider ‘Integration of digital health technology in hemodialysis’. Edit the legend accordingly, e.g., Digital health technology in hemodialysis is conceptually grouped into two main levels. The inner circle….. Note in Figure 1 the abbreviations HD and PROM need to be mentioned in the legend.

Section 2.1.; Lines 107-111; the current DHT incorporated in the hemodialysis machines; are you referring to what is described in 2.1.1 and 2.1.2? If so, indicate that you will review them below and perhaps shorten the paragraph.

Section 2.1.; Lines 112-118; what are these next level DHT? And what are the feedback-controlled algorithms? The readers would want to see specifics and what it means in practice.

Line 156: What did “and enhance patient perception (35, 36)” mean? Please put in brief the study results with respect to patient reports if the studies included those.

Lines 177-182: would temper the criticism to MyTemp study. For example, instead of saying “had notable design flaws”, consider saying had some limitations. The dialysate temperature of 36.5C might not be a limitation (see line 163 indicated usual human core temperature is 36.5C).

Line 190: “Online clearance measurement of dialysis efficiency is vital for ensuring optimal treatment outcomes for hemodialysis patients” Please add specifics, statistical results that support the “vital” statement, e.g., how patient outcomes or average clearance etc differed between treatments with and without online clearance measurement.

Line 207, the end of sentence re DOPPS study needs reference.

Line 209, “These online tools are facing two additional concerns” why “additional”? what were the prior concerns (none were apparent from the preceding paragraphs)?

Line 210, “are not regularly used by caregivers in their workflow processes” It might be too much to expect that caregivers should use or know how to use these devices when even not all professional providers (eg nurses) use or know how to use the device (as it was said in the next sentence).

Lines 239-242, specifics in how the studies affirmed validity, reliability and practicality would help the reader understand the value of the device.

Lines 244-247, consider adding statistical results on the mentioned outcomes of headaches, paradoxical hypertension, fatigue, and IDWG.

Line 249: what does “couple manually managed sodium tools” mean and the sentence does not have a reference.

Subtitle for 2.1.3 is a bit too long

Lines 293-312: the discussion on how QSUB and QD can be modified in HDF to mitigate water consumption might not necessarily fit with the focus of the article on DHT. Might also need to acknowledge that if the studies did not use the adjustments in QSUB and QD considered in this paragraph, the nephrology community will likely not easily embrace this idea.

For the paragraphs of lines 327-342: is this also referring to the cost of designing compatible interface and patient medical data cross-reference between dialysis EMRs and healthcare systems EMRs? If not, consider adding a brief discussion about this, as it is a significant issue for patient care but the costs of reconfiguring large scale EMRs for cross-compatibility – and who would cover these costs – are prohibiting such an achievement.

Figure 2, missing title

Line 428:” The DOPPS study clearly demonstrated that use of best practice patterns” would help to specify what those best practice patterns encompass.

Line 430: “Caregivers should be mindful of this crucial finding,….” casts too much pressure on caregivers. Care for patients treated with dialysis is extremely complex, healthcare systems and insurances do not have nor financially cover caregiver education and time they would need to forego on their other duties to learn, continuously stay educated, and provide that level of care. Hence, the “shortage” of caregivers.

Line 472: would the simulations and obtained results be able to adjust for selection bias, since is using data from patients who, for example, chose home dialysis who inherently had different unmeasurable characteristics that contributed to their outcomes? One might say that presenting patients who are genuinely unable to pursue home dialysis with simulated outcomes from patients possessing cognitive, financial, and social resources for home dialysis could exacerbate the emotional burden on individuals making crucial decisions.

Line 487: would give comparative statistical results and re-add the reference at the end of the sentence.

Other comments which are minor

Reviewer’s personal view: dialysis, ESKD, hemodialysis, CKD to not be used as adjectives in scientific papers. E.g., instead of dialysis patient, or ESKD patient, etc, could write patient on dialysis, patient with ESKD, etc.

Streamline abbreviations that want to be used. Example: at times, terminology is abbreviated and later it re-appears in full writing. E.g., digital health technology appears written in full after its DHT abbreviation was indicated. E.g.,  Line 278: HD and HDF are abbreviated but these have been used before and after that line as unabbreviated.

 Change typo “narrative assay” into “narrative essay”

No need to abbreviate CKD in the Abstract as is used only once

Author Response

A precise, point-to-point response to your comments and concerns has been provided in the attached document.

This is a comprehensive and rather unique review of digital health technology (DHT) in the field of hemodialysis, covering past accomplishments, ongoing explorations, and future opportunities. This Reviewer has a few comments that may help the authors at the revision stage.

Re - Thanks for the comments and suggestions, they will help improving readability of our manuscript

Main comments

In the Abstract and Introduction, outline the method of review, e.g., systematic, scoping; and indicate the field of review, e.g., in-center hemodialysis only, home hemodialysis, hemodialysis and hemodiafiltration, peritoneal dialysis etc.

Regarding sentence in the Abstract and the theme throughout the manuscript: “Managing ESKD requires a shift in treatment paradigms, emphasizing patient education, caregiver knowledge, and robust digital support systems.” Consider rephrasing. This might be viewed as an overstatement. The paradigm of ESKD management should have already included patient and caregiver education: The fact that it has not been done to satisfaction it does not constitute a paradigm shift. However, perhaps DHT will aid in facilitating and enhancing patient and caregiver education.

Re - We agree; the sentence has been adapted accordingly.

Introduction, page 1, line 39, consider changing “and satisfy patient perception” to “and improve patient coping with dialysis treatments”. To “satisfy” patient perception appears a few times throughout the manuscript. Consider rewording, e.g., to ‘mitigate patient and caregiver burden in coping with dialysis’ etc

Re - We agree; the sentence has been changed accordingly

Introduction, page 1, line 42, “and overall poor patient health related quality of life perception worldwide”. Consider removing the word ‘perception’ because poor QoL is a reality rather than just perception.

Re - We agree; Perception was used as a sense by the patient, not as a feeling, since it has been replaced as suggested

Figure 1. is missing a title, consider ‘Integration of digital health technology in hemodialysis’. Edit the legend accordingly, e.g., Digital health technology in hemodialysis is conceptually grouped into two main levels. The inner circle….. Note in Figure 1 the abbreviations HD and PROM need to be mentioned in the legend.

Re - We agree; the title has been added as well as abbreviations

Section 2.1.; Lines 107-111; the current DHT incorporated in the hemodialysis machines; are you referring to what is described in 2.1.1 and 2.1.2? If so, indicate that you will review them below and perhaps shorten the paragraph.

Re - We agree; a change and shorten has been done

Section 2.1.; Lines 112-118; what are these next level DHT? And what are the feedback-controlled algorithms? The readers would want to see specifics and what it means in practice.

Re - We agree; few discussions and examples are given

Line 156: What did “and enhance patient perception (35, 36)” mean? Please put in brief the study results with respect to patient reports if the studies included those.

Re - We agree; little discussion is given

Lines 177-182: would temper the criticism to MyTemp study. For example, instead of saying “had notable design flaws”, consider saying had some limitations. The dialysate temperature of 36.5C might not be a limitation (see line 163 indicated usual human core temperature is 36.5C).

Re - We agree; we simplify the discussion on MyTemp and add an easier explanation

Line 190: “Online clearance measurement of dialysis efficiency is vital for ensuring optimal treatment outcomes for hemodialysis patients” Please add specifics, statistical results that support the “vital” statement, e.g., how patient outcomes or average clearance etc differed between treatments with and without online clearance measurement.

Re - We agree; we provided explanation of why online measurement may improve outcomes

Line 207, the end of sentence re DOPPS study needs reference.

Re - We agree; that has been done

Line 209, “These online tools are facing two additional concerns” why “additional”? what were the prior concerns (none were apparent from the preceding paragraphs)?

Re - We agree; this point has been corrected

Line 210, “are not regularly used by caregivers in their workflow processes” It might be too much to expect that caregivers should use or know how to use these devices when even not all professional providers (eg nurses) use or know how to use the device (as it was said in the next sentence).

Re - We agree; this point has been adapted in a more meaningful manner

Lines 239-242, specifics in how the studies affirmed validity, reliability and practicality would help the reader understand the value of the device.

Re - We agree; some examples are discussed

Lines 244-247, consider adding statistical results on the mentioned outcomes of headaches, paradoxical hypertension, fatigue, and IDWG.

Re - We agree; it is too early to obtain reliable statistics, but some review references are provided

Line 249: what does “couple manually managed sodium tools” mean and the sentence does not have a reference.

Re - We agree; this section has been deleted because the results are too preliminary

Subtitle for 2.1.3 is a bit too long

Re - We agree; the title has been shortened

Lines 293-312: the discussion on how QSUB and QD can be modified in HDF to mitigate water consumption might not necessarily fit with the focus of the article on DHT. Might also need to acknowledge that if the studies did not use the adjustments in QSUB and QD considered in this paragraph, the nephrology community will likely not easily embrace this idea.

Re - We agree; the paragraph has been cut, shortened, and referenced to a recent article discussing the subject

For the paragraphs of lines 327-342: is this also referring to the cost of designing compatible interface and patient medical data cross-reference between dialysis EMRs and healthcare systems EMRs? If not, consider adding a brief discussion about this, as it is a significant issue for patient care but the costs of reconfiguring large scale EMRs for cross-compatibility – and who would cover these costs – are prohibiting such an achievement.

Re - We agree; a sentence on cost-related issue has been added

Figure 2, missing title

Re - We agree; the title has been added

Line 428:” The DOPPS study clearly demonstrated that use of best practice patterns” would help to specify what those best practice patterns encompass.

Re - We agree; a few examples of practices that can be modified have been added

Line 430: “Caregivers should be mindful of this crucial finding,….” casts too much pressure on caregivers. Care for patients treated with dialysis is extremely complex, healthcare systems and insurances do not have nor financially cover caregiver education and time they would need to forego on their other duties to learn, continuously stay educated, and provide that level of care. Hence, the “shortage” of caregivers.

Re - We agree; a few additional comments clarifying the context have been provided

Line 472: would the simulations and obtained results be able to adjust for selection bias, since is using data from patients who, for example, chose home dialysis who inherently had different unmeasurable characteristics that contributed to their outcomes? One might say that presenting patients who are genuinely unable to pursue home dialysis with simulated outcomes from patients possessing cognitive, financial, and social resources for home dialysis could exacerbate the emotional burden on individuals making crucial decisions.

Re - We agree; A few additional comments have been incorporated to discuss this aspect

Line 487: would give comparative statistical results and re-add the reference at the end of the sentence.

Re - We agree; the sentence has been modified accordingly

Other comments which are minor

Reviewer’s personal view: dialysis, ESKD, hemodialysis, CKD to not be used as adjectives in scientific papers. E.g., instead of dialysis patient, or ESKD patient, etc, could write patient on dialysis, patient with ESKD, etc.

Re - We agree; however, it is a common practice to use such adjectives to simplify writing. We have tried to comply as much as possible with this recommendation

Streamline abbreviations that want to be used. Example: at times, terminology is abbreviated and later it re-appears in full writing. E.g., digital health technology appears written in full after its DHT abbreviation was indicated. E.g.,  Line 278: HD and HDF are abbreviated but these have been used before and after that line as unabbreviated.

Re - We agree; we tried to correct it as much as possible, and leave the final correction to the editor

Change typo “narrative assay” into “narrative essay”

Re - We agree; this is corrected

No need to abbreviate CKD in the Abstract as is used only once

Re - We agree; this is corrected

Reviewer 2 Report

Comments and Suggestions for Authors

I have reviewed the manuscript entitled by, Digital Health Support: Current Status and Future Development for Enhancing Dialysis Patient Care and Empowering Patients”.

Authors offer an update on currently available digital health technologies used in the management of hemodialysis patients and envisions future technologies with digital solutions to empower patients.

Major comment:

1. Author has explained the detailed information about the tools. For readers, it would be more convenient to read this information in a table form.

2. Make a table specifying the name of tools and their respective functions.

3. In 2.1.4. subheading, more data should be added for the tools for interfacing dialysis machines and monitoring devices.

4. About biosensors, give details about biosensors (examples).

Minor comment:

The whole manuscript needs to be checked for grammar.

Comments on the Quality of English Language

The whole manuscript needs to be checked for grammar.

Author Response

A precise, point-to-point response to your comments and concerns has been provided in the attached document.

I have reviewed the manuscript entitled by, “Digital Health Support: Current Status and Future Development for Enhancing Dialysis Patient Care and Empowering Patients”.

Authors offer an update on currently available digital health technologies used in the management of hemodialysis patients and envisions future technologies with digital solutions to empower patients.

Re - Thank you for the encouraging and positive comments

Major comment:

  1. Author has explained the detailed information about the tools. For readers, it would be more convenient to read this information in a table form.

Re - We agree; we provided a table summarizing tools and functions

  1. Make a table specifying the name of tools and their respective functions.

Re - We agree; we provided a table summarizing tools and functions

  1. In 2.1.4. subheading, more data should be added for the tools for interfacing dialysis machines and monitoring devices.

Re - We agree; subheading has been added

  1. About biosensors, give details about biosensors (examples).

Re - We agree; a few comments have been added on biosensors currently available or in development

Minor comment:

The whole manuscript needs to be checked for grammar.

It was reviewed by two English-native speaking co-authors

Reviewer 3 Report

Comments and Suggestions for Authors

1) General comments:

The paper addresses current status and future development of digital health support for dialysis patient care and empowering patients.

The paper can be regarded to be of interest for ESRD and CKD community and even wider.

1) General comments and observations.

The review paper is generally very well-written and the content is supported with the relevant data and up-to-date references. The study would be acceptable after a minor revision and the authors should be encouraged to consider a resubmission.

2) Specific revision comments:

Abstract

Page 1, line 10: a comma should be added in “So, alternative strategies, ”.

1. Introduction.

·          Page 2, line 83-84, Chapter “2. Potential of Digital Health Technologies in Dialysis Patient Care”: considering that “electronic scale” and “wearable sensor devices” belong into different categories the text “… utilize wearable sensor devices (such as wristwatch, electronic scale, oximetry, sleep disorder monitoring, and physical activity tracking)” would be more correct when written as “Thirdly, remote tools utilizing electronic scale, wearable sensor devices such as oximetry and wristwatches featured by sleep disorder monitoring and physical activity tracking” or similar.

·          Page 4, line 133: A question is whether subsection titles “Ultrafiltration Controlled System with Profiles, Thermal Balance, and others following” should be numerated to provide a more systematic structure?

·          Page 5, line 181-182: Is the statement “Additionally, it used a much lower dialysate temperature (36.5°C) than previous studies.” justified (e.g. statistically)? Otherwise, it should be written “Additionally, it used lower dialysate temperature (36.5°C) than previous studies.”

·          Page 7, line 272: As a remark that for B2M monitoring in the spent dialysate there is a newer reference available “Paats, J., A. Adoberg, J. Arund, I. Fridolin, K. Lauri, L. Leis, M. Luman and R. Tanner (2021). "Optical Method and Biochemical Source for the Assessment of the Middle-Molecule Uremic Toxin β2-Microglobulin in Spent Dialysate." Toxins 13(4): 255.”

·          Page 7, line 275-277: The chapter title “2.1.3. Tools Designed to Promote More Sustainable Dialysis, with a Specific Emphasis on Reducing Water Consumption: Making the Case for Online Hemodiafiltration and  Adjusting Dialysate Flow in Alignment with Blood Flow” seems too long and should be shortened.

·          Page 8, line 324-326: The sentence “These solutions exemplify the convergence of healthcare and information technology (IT) systems, representing the evolving field known as medical informatics (86).” should be rewritten since there it can be misunderstood as the field medical informatics has arisen recently, but in reality it has a more than 70 years history behind (Collen, M.F., 2006. History of medical informatics: Fifty years in medical informatics. Yearbook of Medical Informatics, 15(01), pp.174-179.).

·          Page 10, line 425-427: The chapter title “3.2. Tools of Continuous Quality Improvement in Patient Management, Both at the Individual  Level and on a Broader Scale at Population Level (e.g., within the Dialysis Unit, Clinic, or  Network)” seems too long and should be shortened.

·          Page 12, line 487-488: By definition it would be more correct to use “Interestingly, it appears that artificial intelligence utilizing machine learning were more accurate than traditional modeling relying on correlations.” than existing “Interestingly, it appears that artificial intelligence and machine learning were more accurate than traditional modeling relying on correlations.” as AI is intrinsically based on various ML algorithms.

·          Page 13, line 522 -528: The section 4.1 is somewhat confusing in the current form as data analysis and secure storing are also technical challenges. It should be rewritten, for example starting to point out first “Technical challenges related to sensors, devices and digital tools are not the most complex to solve, as technology progresses rapidly, providing new solutions every day. …” Later “The primary challenge lies in data handling and management - analyzing the data, providing actionable information to users to facilitate care and decision support, and then securely storing this big data within an appropriate secured cloud system.”

References

·          Please correct the reference (92) as “Shinkman, R., 2018. Is “empowered dialysis” the key to better outcomes?. NEJM Catalyst, 4(2).”

·          Please correct the reference (93) as “Crews, D.C., McCowan, P. and Saffer, T., 2021. Bringing Kidney Care Home: Lessons from Covid-19. NEJM Catalyst Innovations in Care Delivery, 2(2).”

Author Response

A precise, point-to-point response to your comments and concerns has been provided in the attached document.

1) General comments:

The paper addresses current status and future development of digital health support for dialysis patient care and empowering patients.

The paper can be regarded to be of interest for ESRD and CKD community and even wider.

Re- Thanks for the encouraging and positive comment

1) General comments and observations.

The review paper is generally very well-written and the content is supported with the relevant data and up-to-date references. The study would be acceptable after a minor revision and the authors should be encouraged to consider a resubmission.

Thank you for your positive and encouraging comments

2) Specific revision comments:

Abstract

Page 1, line 10: a comma should be added in “So, alternative strategies, ”.

Re - We agree; that has been corrected

  1. Introduction.
  • Page 2, line 83-84, Chapter “2. Potential of Digital Health Technologies in Dialysis Patient Care”: considering that “electronic scale” and “wearable sensor devices” belong into different categories the text “… utilize wearable sensor devices (such as wristwatch, electronic scale, oximetry, sleep disorder monitoring, and physical activity tracking)” would be more correct when written as “Thirdly, remote tools utilizing electronic scale, wearable sensor devices such as oximetry and wristwatches featured by sleep disorder monitoring and physical activity tracking” or similar.

Re - We agree; the sentence has been corrected accordingly

  • Page 4, line 133: A question is whether subsection titles “Ultrafiltration Controlled System with Profiles, Thermal Balance, and others following” should be numerated to provide a more systematic structure?

Re - We agree; subheading numbering has been added for clarity

  • Page 5, line 181-182: Is the statement “Additionally, it used a much lower dialysate temperature (36.5°C) than previous studies.” justified (e.g. statistically)? Otherwise, it should be written “Additionally, it used lower dialysate temperature (36.5°C) than previous studies.”

Re - We agree; this section has been summarized and simplified for clarity

  • Page 7, line 272: As a remark that for B2M monitoring in the spent dialysate there is a newer reference available “Paats, J., A. Adoberg, J. Arund, I. Fridolin, K. Lauri, L. Leis, M. Luman and R. Tanner (2021). "Optical Method and Biochemical Source for the Assessment of the Middle-Molecule Uremic Toxin β2-Microglobulin in Spent Dialysate." Toxins 13(4): 255.”

Re - We agree; the reference has been added

  • Page 7, line 275-277: The chapter title “2.1.3. Tools Designed to Promote More Sustainable Dialysis, with a Specific Emphasis on Reducing Water Consumption: Making the Case for Online Hemodiafiltration and Adjusting Dialysate Flow in Alignment with Blood Flow” seems too long and should be shortened.

Re - We agree; the title has been shortened

  • Page 8, line 324-326: The sentence “These solutions exemplify the convergence of healthcare and information technology (IT) systems, representing the evolving field known as medical informatics (86).” should be rewritten since there it can be misunderstood as the field medical informatics has arisen recently, but in reality it has a more than 70 years history behind (Collen, M.F., 2006. History of medical informatics: Fifty years in medical informatics. Yearbook of Medical Informatics, 15(01), pp.174-179.).

Re - We agree; this point has been integrated in the sentence and the reference added

  • Page 10, line 425-427: The chapter title “3.2. Tools of Continuous Quality Improvement in Patient Management, Both at the Individual Level and on a Broader Scale at Population Level (e.g., within the Dialysis Unit, Clinic, or  Network)” seems too long and should be shortened.

Re - We agree; the title has been shortened accordingly

  • Page 12, line 487-488: By definition it would be more correct to use “Interestingly, it appears that artificial intelligence utilizing machine learning were more accurate than traditional modeling relying on correlations.” than existing “Interestingly, it appears that artificial intelligence and machine learning were more accurate than traditional modeling relying on correlations.” as AI is intrinsically based on various ML algorithms.

Re - We agree; this has been corrected in the sentence

  • Page 13, line 522 -528: The section 4.1 is somewhat confusing in the current form as data analysis and secure storing are also technical challenges. It should be rewritten, for example starting to point out first “Technical challenges related to sensors, devices and digital tools are not the most complex to solve, as technology progresses rapidly, providing new solutions every day. …” Later “The primary challenge lies in data handling and management - analyzing the data, providing actionable information to users to facilitate care and decision support, and then securely storing this big data within an appropriate secured cloud system.”

Re - We agree; the paragraph has been restructured as suggested to provide more clarity

References

  • Please correct the reference (92) as “Shinkman, R., 2018. Is “empowered dialysis” the key to better outcomes?. NEJM Catalyst, 4(2).”

Re - We agree; this is done

  • Please correct the reference (93) as “Crews, D.C., McCowan, P. and Saffer, T., 2021. Bringing Kidney Care Home: Lessons from Covid-19. NEJM Catalyst Innovations in Care Delivery, 2(2).”

Re - We agree; this is done

Round 2

Reviewer 2 Report

Comments and Suggestions for Authors

The comments given for the improvement of the manuscript have been addressed by authors. 

Now, the manuscript is eye-catchy and is in the acceptable form.